# Novel Synthesis Methods of New Imidazole-Containing Coordination Compounds Tc(IV, V, VII)—Reaction Mechanism, Xrd and Hirshfeld Surface Analysis

**DOI:** 10.3390/ijms23169461

**Published:** 2022-08-21

**Authors:** Mikhail Alexandrovich Volkov, Anton Petrovich Novikov, Mikhail Semenovich Grigoriev, Alexander Mikhailovich Fedoseev, Konstantin Eduardovich German

**Affiliations:** Frumkin Institute of Physical Chemistry and Electrochemistry, Russian Academy of Sciences, Leninskii Prospect 31-4, 119071 Moscow, Russia

**Keywords:** technetium, imidazole, technetyl, XRD-analysis, Hirshfeld surface analysis, π-interactions, metal-nitrogen bond, supramolecular chemistry

## Abstract

In this work, we have proposed two new methods for the synthesis of [TcO_2_L_4_]^+^ (where L = imidazole (Im), methylimidazole (MeIm)) complexes using thiourea (Tu) and Sn(II) as the reducing agents. The main and by-products of the reactions were determined, and possible reaction mechanisms were proposed. We have shown that the reduction of Tc(VII) with thiourea is accompanied by the formation of the Tc(III) intermediate and further oxidation to Tc(V). The reaction conditions’ changing can lead to the formation of Tc(VII) and Tc(IV) salts. Seven new crystal structures are described in this work: Tc(V) complexes, salts with Tc(VII) and Tc(IV) anions. For the halide salts of Tu the cell parameters were determined. In all of the obtained compounds, except for [TcO_2_(MeIm)_4_]TcO_4_, there are π–stacking interactions between the aromatic rings. An increase in the anion size lead to weakening of the intermolecular interactions. The halogen bonds and anion-π interactions were also found in the hexahalide-containing compounds. The Hirshfeld surface analysis showed that the main contribution to the crystal packing is created by the van der Waals interactions of the H···H type (42.5–55.1%), H···C/C···H (17.7–21.3%) and hydrogen bonds, which contribute 15.7–25.3% in total.

## 1. Introduction

Technetium is mostly chemically and environmentally available as Tc(VII)O_4_^–^, and therefore in most reactions possesses oxidative properties. Depending on the reducing conditions and the chemical nature of the ligand available, Tc(VII)O_4_^–^ may convert to stable/unstable final/intermediate species of Tc(V), Tc(VI) or Tc(III). Under very special conditions, the products could be either of a cluster nature [1], or carbonyl derivatives [2,3] of Tc[2+, 2.5+ and 0] oxidation states. Very high attention is focused on the Tc(V) and Tc(III) complexes that play important roles in the preparative [4,5,6], radiopharmaceutical [7,8] and industrial reprocessing chemistry of Tc [9]. 

Weak, intermolecular interactions are of particular interest from a theoretical point of view, the study of which in pertechnetates and perrhenates of purines led to the discovery of a new type of chemical bond [10]. New methods of separating perrhenates and pertechnetates are based on non-valent interactions [11,12,13,14]. The study of the non-valent interactions’ contribution for various technetium compounds will allow a deeper understanding of the impact of radiopharmaceuticals on living organisms [7], and of the formation of non-trivial technetium complex compounds, similar to those described in [4,5,15]. The information about weak interactions can also be useful for predicting the behavior of the Tc-containing intermediates in catalytic and sorption processes on complex organic sorbents, such as those described [16].

Halide systems have now received a new meaning for high-temperature pyrometallurgical reprocessing of spent nuclear fuel. One approach to this is the use of low-temperature melts and ionic liquids, based on imidazole derivatives. It is known that concentrated hydrohalogen acids form the stable compounds TcO(Hal)_4_^2–^, but during its long-term storage, Tc(V) transforms into the hexahalide compounds of Tc(IV). However, with the presence of ligands in the Tc(V) solutions, it becomes possible to obtain useful functional complexes, which was shown in the work [17]. There is no description in the literature of Tc(IV) hexahalide complexes with imidazolium cations. In the course of the operation of molten-salt reactors, the TcHal_6_^2–^ compounds will inevitably accumulate. The study of the crystal structure, intermolecular interactions and conditions for the formation of technetium hexahalides can be used to develop methods for reprocessing fuel for new types of reactors [18,19]. It should be taken into account that TcHal_6_^2–^ compounds can be used to purify and obtain metallic technetium, which is considered to be the most acceptable form for the long-term storage or transmutation of technetium. 

The reduction of the pertechnetate ions with organosulfur compounds (inorganic Na_2_S_2_O_3_ or organic thiourea (Tu)) leads to the formation of Tc(V) [20,21] depending on the present ligands, which the authors of the following work pay attention to [22].

In the work of Fackler and co-workers in 1985 [1], the structure of [TcO_2_(Im)_4_]Cl·2H_2_O (Im = imidazole) was analyzed. The Tc=O distances were used in some of the databases as a reference for TcO_2_^+^, giving an idea of the large variation of Tc=O in technetyl cation—the observation being in fact erroneous, due to the initial misinterpretation of the space group in the compound structure. Meanwhile, the great interest in the technetyl(V) derivatives arose due to the fact that its application in nuclear medicine as biologically active ^99m^Tc radiopharmaceuticals is based on the Tc(V) formation with [O=Tc=O]^+^ as a complex-forming center [23,24,25,26]. The Tc complexes of such a type with amine and imine ligands are the principal models for the Tc bonding to amino acids, proteins, antibodies etc. In general, Tc (V) usually forms trans -[O_2_L_4_Tc]^+^, where L is the N-bearing organic ligand [17]. Imidazole rings are often found in biochemical systems applicable to metal coordination.

The R-factor obtained in [17] for [TcO_2_(Im)_4_]Cl·2H_2_O, was equal to 0.091 with the abnormally short distances Tc–O, that are characteristic more of Tc(VII) than of Tc(V). The analyses of local symmetry and the *hkl* set support the erroneous interpretation of the space group to a centrosymmetric one by the authors of [17] for the structure [TcO_2_(Im)_4_]Cl·2H_2_O. Nowadays, a great database for metalloyl cations complexes is available, and the typical tendencies indicate that the structure of [TcO_2_(Im)_4_]Cl·2H_2_O in [17] was confusing; therefore, we considered it reasonable to undertake new research of its analogues with the principal idea of enlarging (and possibly correcting) the database for the technetyl complex compounds and the structural understanding of them.

The paper considers a new method for the preparation of [TcO_2_(Im)_4_]Cl·2H_2_O (**I**), [TcO_2_(Im)_4_]Br·2H_2_O (**II**), [TcO_2_(2-MeIm)_4_]Cl·2H_2_O (**III**), and [TcO_2_(2-MeIm)_4_]TcO_4_ (**IV**) compounds, their crystal structures are studied and the noncovalent interactions are analyzed by the Hirshfeld surfaces method and IR spectroscopy of [TcO_2_(Im)_4_]Br·2H_2_O. A detailed analysis of the reactional by-products ((MeIm)(HMeIm)(Tu)TcO_4_ (**V**), HIm_2_TcCl_6_ (**VI**), Him_2_TcBr_6_ (**VII**), (HMeIm)_4_Tc/SnCl_6_Cl_2_ (**VIII**), Tu_2_Cl_2_ (**IX**) and Tu_2_Br_2_ (**X**)) was carried out and new imidazole complexes Tc(VII) and Tc(IV) were identified. The optimal conditions for the formation of the complex-forming nucleus TcO_2_^+^ by the proposed method were also determined.

## 2. Results and Discussion

### 2.1. Structural Description of Tc(V) Complexes

Compounds **I**–**IV** contain [TcO_2_L_4_]^+^ cationic complexes, in which the Im (MeIm) molecules are coordinated to the Tc atom by the N atom (Figure 1). The crystal structure data for compound **I** were obtained from [17]. The [TcO_2_Im_4_]Cl·2H_2_O structure described earlier in the centrosymmetric space group *C2/c*, with the disordering of the chloride ion and water molecules is, probably, not centrosymmetric and similar to **II**. Furthermore, in [17] the authors also reported the structure of [TcO_2_(MeIm)_4_]Cl_2_H_2_O. The space group of the compound is *C*2/*c*, while the compound **III** of the same composition in our work is *P*-1. The oxygen atoms are in the trans-position (the angles O-Tc-O are close to 180°) The N-Tc-N angles are close to 90° in all of the complexes (Appendix A). In **II**, three imidazole rings are turned in one direction (C1, C7 and C10 atoms on one side of the equatorial plane of the TcO_2_ group), and only one ring is turned in the other direction (C4 on the other side of the plane) (Figure 1). In the structures **III** and **IV**, the opposite imidazole rings are turned in different directions. Although in all of the structures the imidazole rings are planar, they are not parallel to the O···O direction. In structure **II**, the two opposite rings are rotated further away from this direction (torsion angles O1-Tc1-N1-C3 and O1-Tc1-N3-C4 are 159.1° and 164.2°, respectively), than the other two (the torsion angles O1-Tc1-N7-C12 and O1-Tc1-N5-C8 are 170.1° and 175.4°). In **III**, the imidazole rings deviate more strongly from the O···O direction (torsion angles change from 150° to 155°), while in **IV**, on the contrary, they are close to this direction (torsion angles change from 168.1° to 175.0°).

The Tc–N distances in all of the complexes are approximately equal and change from 2.122 to 2.140 Å. The Tc=O distances in the complexes vary from 1.743 to 1.763 Å, which is longer than the distances described in previous works for the chlorides of these complexes [17], and used in the calculations [27]. 

In all of the structures, the cations are bonded to each other through crystallization water molecules or counterions. In **II**, the cations are linked through bromide-ions by the H-bonds of the N–H···Br type and through the water molecules by the hydrogen bonds of the N–H···O and O–H···O types. In **III**, the cations are linked to the water molecules by the hydrogen bonds of the O–H···O type, which in turn form a water···Cl···water···water bridge to another cation. In **IV**, the cations are linked to each other through the pertechnetate anions by the weak hydrogen bonds of the C–H···O type. The crystal packing in all of the complexes can be represented as layered, where the molecules of the complexes form layers, between which there are molecules of anions and water (Figure 2). Additionally, the molecules of the complexes in structure **II** are connected by two π–stacking interactions, and in **III** by only one (Figure 3a,b). The strongest π–stacking interaction is observed in structure **III** (Table 1). In structure **IV**, there are short distances between the atoms of the imidazole rings, but these contacts cannot be called π–stacking [28,29], because of the large distance between the centers (4.127 Å) and the large displacement (1.944 Å) of the rings. However, in **IV**, the methyl hydrogen of methylimidazole and one of the hydrogens of the imidazole ring participate in the CH-π interactions (the distances between the hydrogen atoms and the centers of the aromatic rings are shorter than 3 Å(Figure 3c)) [30,31,32,33]. In this case, the distance between the hydrogen atom and the center of the aromatic ring is the shortest in the case of the hydrogen of the imidazole ring. Although the CH-π interactions are often considered weaker than the hydrogen bonds, they are one of the factors that determine the molecular structure within a crystal and can be used for recognition [34,35].

The analysis of the supramolecular interactions in the obtained structures showed that an increase in the size of the anion leads to a deterioration in the binding of the molecules in the crystal but does not lead to crystal-packing disturbances (solvent molecules and counterions are located between the layer of cations) and suggests an increase in solubility upon passing from compound **II** to **IV**.

### 2.2. Structural Description of By-Products

Compound **V** was obtained as a by-product of the preparation of compound **IV**. The structure contains two MeIm molecules, one of which is protonated at the nitrogen atom, but the proton in the compound is disordered between the two nitrogen atoms of different molecules (N3 and N13). The structure contains a neutral thiourea molecule and a pertechnetate-anion (Figure 4a). The MeIm fragments are in the same plane (the angle between the planes is 4.67°). The crystal packing can be represented as layered, the MeIm fragments are linked to each other by a hydrogen bond of the N–H···H type into dimers and by π–stacking interaction (Table 2) into layers, as in **II**–**IV**, and the pertechnetate-ions and thiourea molecules are located between the layers (Figure 4b). The thiourea molecules are linked to each other by the hydrogen bonds of the N–H···S type, to the pertechnetate-ions by H-bonds of the N–H···O type and to the imidazole fragments by the bonds of the C–H···S type.

Compounds **VI**, **VII** and **VIII** are by-products of the second method. Compounds **VI** and **VII** contain protonated imidazole cations and hexahalide Tc(IV) anions (Figure 5). Compound **VIII** is a mixed salt containing 30% technetium and 70% tin in the hexahalide anion. In **VIII**, the asymmetric fragment contains four MeIm cations, the hexahalide anion and two chloride anions. The environment of the technetium or tin atoms in the hexahalide anions is close to an ideal octahedral (Appendix A).

In structures **VI–VIII**, there are π–stacking interactions between cations (Table 2) (Figure 6). The crystal packing can be represented as layered. The cations in **VI**–**VIII** are linked into layers by the π–stacking interactions, the cations and anions are linked between layers by hydrogen bonds (Figure 7). In **VI** and **VII**, the anions are additionally linked by the halogen bonds, and the Hal···Hal. The packing of **VI** and **VII** is very close (Figure 7), the main difference is in the small rotations of anions and cations: in **VI** (sp. gr. C2/*m*), the anions occupy a special position with symmetry 2/*m*, and cations occupy a special position with symmetry *m*; in **VII** (sp. gr. *P*2_1_/*c*), the anions occupy a special position with symmetry -1, and the cations are in a general position. 

In addition, in **VI**–**VIII** there are anion–π interactions (the distance between the center of the ring and the anion is less than 5 Å, and the angle α is greater than 50°) between the hexahalide anions and imidazole rings (Figure 6) [36,37].

For compound **IX**, a crystal structure containing a dithiourea cation and chloride anions was previously described [38]. The doubly charged dithiourea cation is formed by S–S bonding of the thiourea molecules. The halide anions are attached to the cations by the hydrogen bonds. Compound **X** is isostructural to compound **IX** (a = 8.836(1), b = 10.506(1), c = 19.695(1), sp. gr. *Pbca*); in this paper, we determined only the crystal cell of compound **IX**: a = 9.36, b = 10.70, c = 20.17 Å, orthorhombic *P*.

### 2.3. Hirshfeld Surface Analysis

The Hirshfeld surface (HS) analysis is based on the division of the electron density in a crystal. The Hirshfeld surface covers the molecule and determines the volume of space in which the electron density of the pro-molecule exceeds the density of all of the neighboring molecules [39]. The fingerprint plots (2D scans of 3D surfaces) are a convenient way to analyze the intermolecular interactions present in crystals. This method can be used to analyze the π–stacking interactions [40], the halogen and hydrogen bonds [41,42], anion–π [28] and other weak non-covalent interactions [43,44].

The Crystal Explorer 21 [45] program was used to analyze the non-valent interactions in crystals using the HS analysis. The donor–acceptor groups are visualized using a standard (high) surface resolution and *d*_norm_ surfaces (Figure 8a–c). The red spots on the surface of the *d*_norm_ plot indicate intermolecular contacts involving the hydrogen bonds. The brightest red spots correspond to the strongest hydrogen bond N—H···Hal in **II–III** and O–H···O in **II**. The weaker red spots correspond to C—H···Hal bonds in **II**–**III**, C—H···O in **II**–**IV** and the π–stacking interactions in **II**–**III**. For an additional analysis of the π–stacking interactions, the shape-index surfaces were described. There are π–stacking interactions in **II** and **III**, as seen from the characteristic red and blue triangles on the surface, and absent in **IV** (Figure 8d–f). 

The nonvalent interactions analysis of the obtained complexes showed that the main contribution to the crystal packing is made by van der Waals interactions of the H···H type (42.5–55.1%) and H···C/C···H (17.7–21.3%). The contacts of the H···H type make the greatest contribution to **III**, which may be due to the appearance of a methyl group and the absence of a large number of H-bonds. A significant contribution to the intermolecular interactions is made by the hydrogen bonds, for which the following types are responsible: O···H/H···O and Hal···H/H···Hal, which contribute 15.7–25.3% in total. Contacts of the C···N/N···C type responsible for the π–stacking interactions contribute 3.3% in **II**, 1.5% in **III** and are practically absent in **IV** (0.1%). The contacts that contribute less than 2% are not considered in Figure 9.

### 2.4. Ir-Spectroscopy

The electronic absorption spectroscopy of the bright pink solutions of compounds **I** and **II** showed a broad wave with a peak of 480 ± 10 nm (Figure 10, inset), which is typical for Tc(V) compounds. Otherwise, the UV-Vis spectroscopy was uninformative.

As no sulfur atoms are present in our case, compound **II**, the group [O=Tc=O]^+^ is completely characterized with the fluctuation bands at 790–880 cm^−1^ in the infrared region [46,47,48]. In the IR spectrum of **II** (Figure 10) the Tc=O group is pronounced by the bands at 809.79 cm^−1^ and 847.68 cm^−1^ (Table 2). A wide wave in the range of 2800–3700 cm^−1^ corresponds to the vibrations of water atoms [49] and 1500 ± 50 cm^−1^ scissor vibrations HOH. The remaining peaks displayed on the spectrum, including those that appear in the region of 2800–3700 cm^−1^, correspond to the vibrations of the carbon and nitrogen atoms in the imidazole fragment of the compound molecule **II** [50,51]. The spectra show the peaks splitting typical for compressed tablets KBr.

### 2.5. Proposed Mechanism for the Formation of Complexes [Tco_2_(Im)_4_]^+^

The reduction reaction of the pertechnetate ions with thiourea in the presence of the HHal acids has a relatively low yield, but it also has advantages. The method turned out to be selective for the synthesis of the [TcO_2_(R-Im)_4_]^+^ complexes, and the target Tc-containing products were the only colored compounds. Changing the concentrations of thiourea and hydrohalic acid did not lead to a noticeable change in the yield of the target reaction products. Nevertheless, it should be noted that, in the absence of the acids (HHal or HTcO_4_), the reaction proceeds very slowly, which can be explained as in the described compound V, where the thiourea interacts with oxygen Tc=O with unprotonated nitrogen atoms of nitrogen. The experiments have shown that an excess of thiourea and an equimolar concentration of HHal acid will be optimal for the preparation of the [TcO_2_(R-Im)_4_]^+^ compounds. The UV-VIS spectroscopy of the intensely colored solutions formed during the synthesis showed the presence of several broad wavelengths of 370 nm, 415 nm and 495 nm (Figure 11). A similar spectrum is described by the authors of [21] (peaks at 423 nm and 493 nm), which suggests the formation of Tc(III) intermediates in the form of coordination cations [TcTu_4_L_2_]^3+^ (L = Tu or R-Im). The authors of this work find it difficult to interpret the peak at 370 nm. The synthesis in an oxygen-free atmosphere hindered the formation of the target complexes; the Tc(III) compounds could not be isolated in the presence of imidazoles. The combination of the above factors helped to suggest the mechanism for the formation of Tc(V) complexes, which is schematically presented in Figure 12.

The reduction of the pertechnetate ion and the formation of the complexes apparently occur during the successive reduction of Tc(VII) to Tc(III) and the subsequent oxidation with atmospheric oxygen to Tc(V). The reaction takes place in the presence of catalytic anions in an acidic medium, in which the thiol tautomeric form of thiourea is stable. The reaction begins with the SH-group reaction with the Tc-O^–^ bond; in this case, the Tc-S bond is formed with the formation of particle I and the water molecule elimination. The redistribution of the charges in an acid medium leads to an elongation of the Tc=O bond and the formation of particle II. Repeating this process three times results in the formation of particle III. The resulting fragment is reactive and coordinates with the two additional ligands, which can be assumed to contain thiourea, in the case of thiourea excess. The resulting particle IV presumably has an octahedral environment of the technetium atom [TcS_6_]. In the presence of the imidazolium fragment, it is possible for a slow stage of fragment IV oxidation. The oxidation, most likely with dissolved oxygen, leads to the sequential elimination of two thiourea molecules, which leave as a Tu_2_^2+^ dimer, and the formation of a V particle. The repeated course of the reaction of oxidation and the substitution of two molecules of the Tu leads to the formation of particle VI. The resulting particle VI coordinates with the two imidazole molecules and completes the coordination sphere, forming a stable reaction product [TcO_2_(Im)_4_]^+^.

The mechanism of the Tc(V) complex formation reaction, according to the second proposed method, is based on the assumption of the formation of an intermediate TcO(Hal)_5_^2–^ particle. The ongoing hydrolysis processes lead to the formation of the [O=Tc=O]^+^ nucleus, and in the presence of an imidazolium ligand, the final product [TcO_2_(Im)_4_]^+^ is formed. A large excess of HHal acid provokes a competitive exchange of ligands; at a sufficiently high concentration, HHal reduces Tc(VII) via Tc(V) to Tc(IV) with the formation of a stable hexahalide complex.

## 3. Materials and Methods

*Caution!* ^99^Tc is a β-emitter (*A* = 635 Bq/μg [52], *E_max_* = 290 keV), the appropriate shielding and manipulation techniques were employed during the synthesis and all of the manipulations.

All of the reagents used in the work were qualified chemically pure and were not subjected to further purification. The technetic acid used in the work was prepared by dissolving Tc_2_O_7_ in bi-distilled water (R ≥ 18 MΩ).

### 3.1. Synthesis [TcO_2_(Im)_4_]Cl·2H_2_O (I), [TcO_2_(Im)_4_]Br·2H_2_O (II), [TcO_2_(2-MeIm)_4_]Cl·2H_2_O (III)

#### 3.1.1. Method 1

A total of 10 mg thiourea (Tu) (Merck) and 10 mg imidazole (Im) (Merck) (5 µL 2-methylimidazole (MeIm) (Merck) for compound **III**) were placed in a 1.5 mL vial and dissolved in 350 µL methanol (Sigma-Aldrich). Then, 10 µL of 12M HCl (5M HBr for compound **II**) and 2 µL of 3.5 M HTcO_4_ were added to the resulting solution. The vial with an intensely colored red solution was closed and left until the color changed to bright pink, then the solution was evaporated to dryness at room temperature (about 10 h). The resulting crystalline mixtures were washed once with cold methanol (3 °C), the washing solutions were colored light pink. The resulting crystals **I**, **II** and **III** were suitable for SCXRD. The compounds are infinitely soluble in alcohol, acetone, water and organic acids. The recrystallization is conveniently carried out from methanol. The approximate reaction yield of each substance by technetium: **I**, **II** ~20%, **III** ~40%. UV-Vis spectroscopy of the methanol solutions of compounds **I**, **II** and **III**: 233 nm, 340 nm, 480 nm. A total of 50 mg of each of the products was given for elemental analysis. The elemental analysis calculated/found (%). **I**: Tc–20.36/19.54; O–13.48/16.17; C–30.34/32.37; N–23.6/24.25; Cl–7.48/7.33; H–4.21; **II**: Tc–19.07/18.7; O–12.33/13.3; C–27.7/29.47; N–21.6/21.76; Br–15.41/14.48; H– 3.9. **III**: Tc–18.66/17.28; O–12.06/13.2; C–36.19/37.63; N–21.11/21.9; Cl–6.7/6.29; H–4.52. 

#### 3.1.2. Method 2

The compounds **I**, **II** and **III** were prepared by reduction of 100 mg KTcO_4_ («Isotope JSC») with SnCl_2_ (Merck) (or SnBr2 for substance II) in 1M HCl (HBr) in presence of excess Im (MeIm for substance **III**). A pink solution was formed on refluxing the reaction mixture at 80 ^o^C for 3 h. The pink crystals were grown on cooling and separated by filtration and recrystallized for ethanol. The mother liquors were an intense red-brown color. The obtained compounds **I**, **II** and **III** turned out to be contaminated with other technetium-containing products and were subjected to recrystallization from methanol. The approximate reaction yield by technetium: **I**, **II** = 50 %, **III** = 63%. 

### 3.2. Synthesis [TcO_2_(2-MeIm)_4_]TcO_4_ (IV)

A total of 9.5 mg of thiourea (Tu) and 5 µL of 2-methylimidazole (MeIm) were placed in a 1.5 mL vial and dissolved in 350 µL of methanol. Then, 5 µL of 12M HCl and 10 µL of 3.5M HTcO_4_ were added to the resulting solution. The intensely colored red solution was allowed to evaporate to dryness at room temperature (10 h). The pink transparent crystals of **IV** were separated from the mother liquor and washed with methanol (20 °C), until the washings were colorless. The resulting compound **IV** was less soluble in the alcohols and acetone than those obtained for **I**, **II** and **III**. Compound **IV** can be conveniently recrystallized from ethanol with the addition of 10% water. Approximate reaction yield by technetium was 40%. A total of 50 mg of the product **IV** was given for elemental analysis. The elemental analysis calculated/found (%): Tc-33.11/31.12; O–6.06/17.5; C–32.1/32.37; N–18.72/19.66; H–4.01.

### 3.3. Selection and Synthesis (MeIm)(HMeIm)(Tu)TcO_4_ (V)

The alcohol solutions formed by washing the dry crystalline mixture of compound **IV** were evaporated by half at room temperature. The transparent colorless crystals that formed along the meniscus of the liquid were selected for X-ray diffraction. An increase in the concentration of MeIm and a decrease in the acidity of the reaction mass led to an increase in the content of compound **V** in the products (approx. 40–50% by Tc). 

### 3.4. Selection and Synthesis HIm_2_TcCl_6_ (VI), Him_2_TcBr_6_ (VII) (HMeIm)_4_Tc/SnCl_6_Cl_2_ (VIII)

The methanolic solutions obtained after washing the mother crystalline mixtures of compounds **I**, **II** and **III** were mixed with the solutions obtained after recrystallization of the compounds obtained by method 2. After the slow evaporation of methanol, the crystalline mixtures were obtained containing a small amount of compounds **I**, **II** and **III** (approx. 5%), compounds **VI**, **VII** and **VIII** (approx. 30%). In a mixture containing compound **II**, the previously described [53] HImBr crystals were found.

Compounds **VI** and **VII** can be obtained analogously to the second synthesis method **I**, **II** and **III** by adding more HHal acid. The recrystallization of the products **I**, **II** and **III** from the concentrated HHal acids also led to the formation of hexahalotechnetates.

### 3.5. Selection Tu_2_Cl_2_ (IX) and Tu_2_Br_2_ (X)

In the dried crystalline mixture containing crystals of compounds **IV** and **V**, as well as in the mixtures of the compounds **I**, **II** and **III**, obtained by method 1, long transparent crystals of **IX** and **X** were found. They were readily soluble in methanol and could be purified from other impurities, including technetium compounds by recrystallization. 

### 3.6. FTIR-Analysis

The IR spectrum of the compound **II** was registered at Nicolet IR200 FT-IR from the 2 mg sample pressed as a finely grounded mixture with 100 mg KBr and pressed at 50 kg/sm^2^.

### 3.7. Single-Crystal XRD Analysis

The crystal structure of all of the synthesized substances was determined by X-ray structural analysis, using an automatic four-circle area-detector diffractometer Bruker KAPPA APEX II with MoKα radiation. The cell parameters were refined over the entire dataset together with the data reduction by using SAINT-Plus software [54]. The absorption corrections were introduced using the SADABS program [55]. The structures were solved by using the SHELXT-2018/2 program [56] and refined by full-matrix least squares on *F^2^* in the anisotropic approximation for all of the non-hydrogen atoms (SHELXL-2018/3 [57]). Atoms H, bounded to CH- and NH-groups, were placed in geometrically calculated positions with the isotropic temperature factors equal to 1.2 *U_eq_*(C, N) and 1.5 *U_eq_*(C) for CH_3_-groups. The H atoms in the water molecules in **I** were objectively located from the difference Fourier synthesis and refined with isotropic temperature factors equal 1.5 *U_eq_*(O). The structure I was refined as an inversion twin. The tables and figures for the structures were generated using Olex2 [58].

The crystal data, data collection and structure refinement details are summarized in Table 3. All of the other crystallographic parameters of the structures are indicated in Appendix A. The atomic coordinates were deposited at the Cambridge Crystallographic Data Centre [59], CCDC № 1022420, 2193592–2193597 for **II**–**VIII**. The supplementary crystallographic data can be obtained free of charge from the Cambridge Crystallographic Data Centre via www.ccdc.cam.ac.uk/data_request/cif (accessed on 30 July 2022).

### 3.8. Elemental Analysis

For compounds **I**–**IV**, the chemical composition was determined on a sample of 50 mg each. The technetium in the compounds was determined by liquid scintillation on a Tri-Carb 3180 TR/SL instrument (PerkinElmer, Chiba, Japan), using a HiSafe 3 scintillator; the measurement error did not exceed 5%. C, N, O were determined using the EA 3000 EuroVector analyzer, the measurement error was not more than 10%. The halogens were determined by Mohr titration in the presence of potassium dichromate.

## 4. Conclusions

In this work, we proposed two new methods for the preparation of [TcO_2_L_4_]^+^ complexes (where L = imidazole (Im), methylimidazole (MeIm)). The reduction of Tc(VII) in an acidic alcoholic solution with thiourea leads to the selective formation of Tc(V). In the absence of a reducing agent, a complex salt Tc(VII) is formed. The reduction of Tc(VII) in an acidic aqueous solution by Sn(II) leads to the formation of a mixture of Tc(V) and Tc(IV) complexes. In a large excess of the reducing agent, the Tc(IV) hexahalides are formed and the formation of double, mixed salt (HMeIm)_4_[Tc/SnCl_6_]Cl_2_ is possible. The main by-products of the reactions are determined and the possible reaction mechanisms are considered. We have shown that the Tc(VII) reduction by thiourea occurs via Tc(III) intermediates, followed by oxidation to Tc(V).

The definition of the [TcO_2_Im_4_]Br·2H_2_O crystal structure in the non-centrosymmetric space group suggests that the literature description of the [TcO_2_Im_4_]Cl·2H_2_O structure in the *C2/c* space group with disordered Cl and water positions may be erroneous. The lengths of the Tc-N and Tc=O bonds in complexes of the [TcO_2_L_4_]^+^ type were refined. A comparative analysis of the non-valent interactions in the obtained crystals showed that with an increase in the size of the anion or the change of Im for MeIm, the intermolecular interactions become weaker. In all of the compounds obtained, except for [TcO_2_(MeIm)_4_]TcO_4_, there are π–stacking interactions between the aromatic rings. The halogen bonds and anion–π interactions were also found in the hexahalide-containing compounds. An analysis of the Hirshfeld surface showed that the main contribution to the crystal packing is made by van der Waals interactions of the type H···H (42.5–55.1%), H···C/C···H (17.7–21.3%) and hydrogen bonds, which contribute 15.7–25.3% in total.

## Figures and Tables

**Figure 1 ijms-23-09461-f001:**
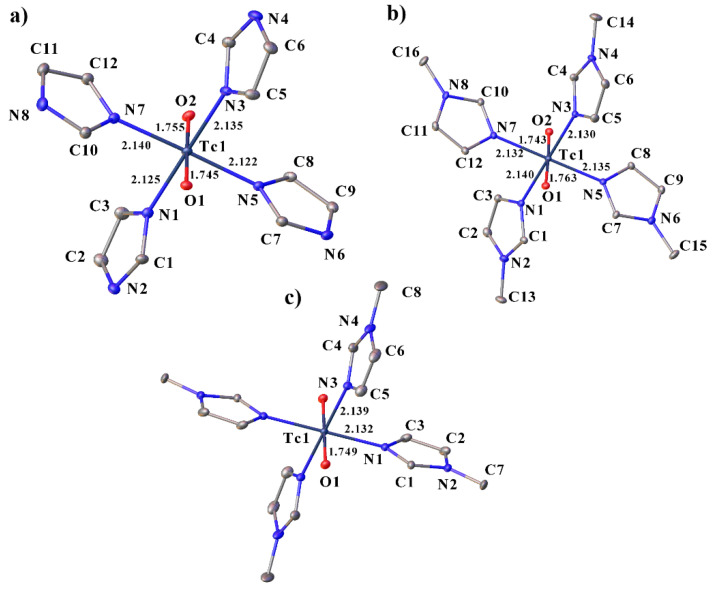
Molecular structure of **II** (**a**), **III** (**b**) and **IV** (**c**) showing the coordination of the technetium atom with some bond lengths and labeling. Solvent molecules, counterions and H-atoms are omitted for clarity.

**Figure 2 ijms-23-09461-f002:**
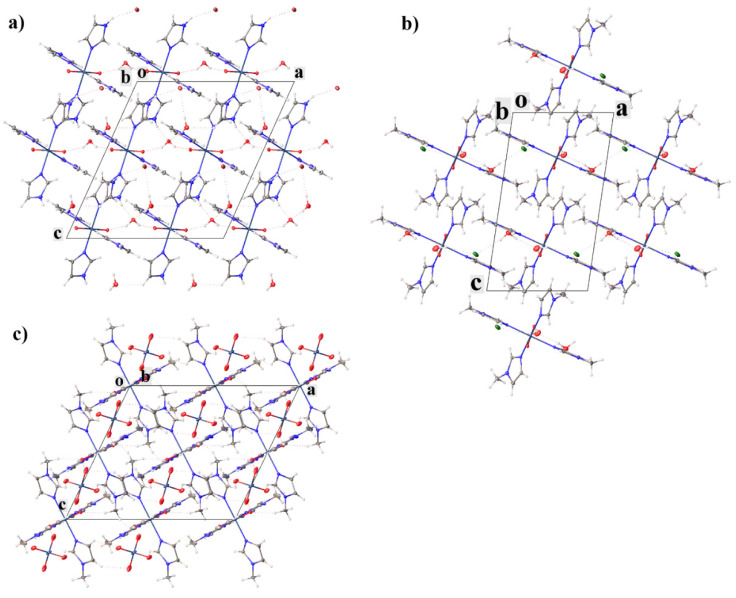
Crystal packing of **II** (**a**), **III** (**b**) and **IV** (**c**) showing layers of complexes with solvent molecules and counterions between them. View along *b* axis.

**Figure 3 ijms-23-09461-f003:**
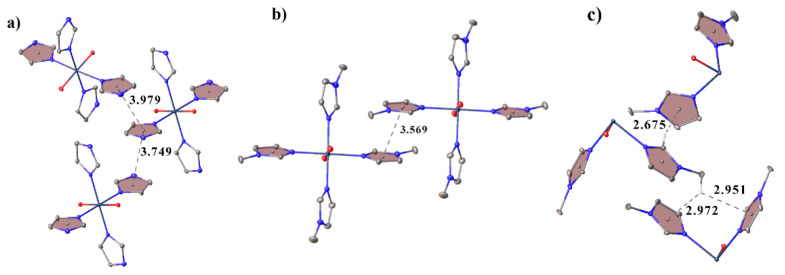
View showing π–stacking interactions in the structures **II** (**a**), **III** (**b**) and CH-π interactions in **IV**(**c**). Only the hydrogen atoms involved in the CH-π interactions are shown.

**Figure 4 ijms-23-09461-f004:**
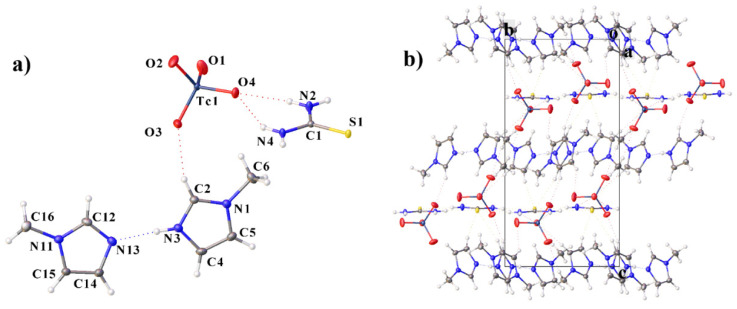
Molecular structure of **V** (**a**) including atom labeling and crystal packing (**b**). Only one disordered H-atom of the imidazole ring is shown.

**Figure 5 ijms-23-09461-f005:**
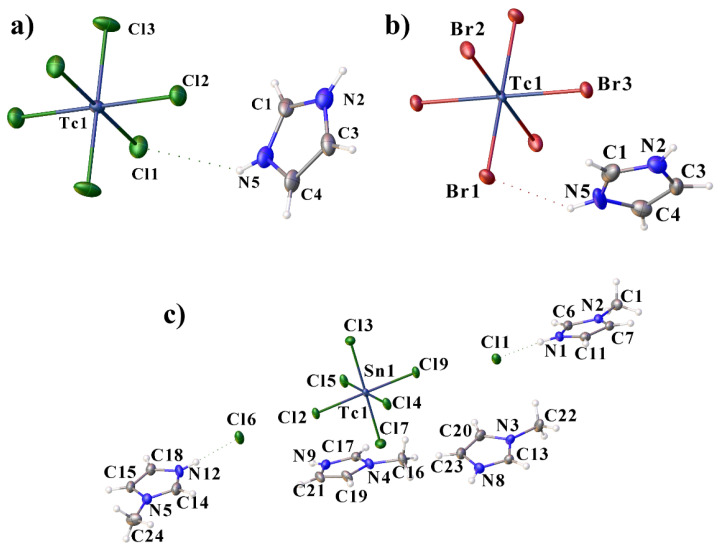
Molecular structure of **VI** (**a**)**, VII** (**b**) and **VIII** (**c**).

**Figure 6 ijms-23-09461-f006:**
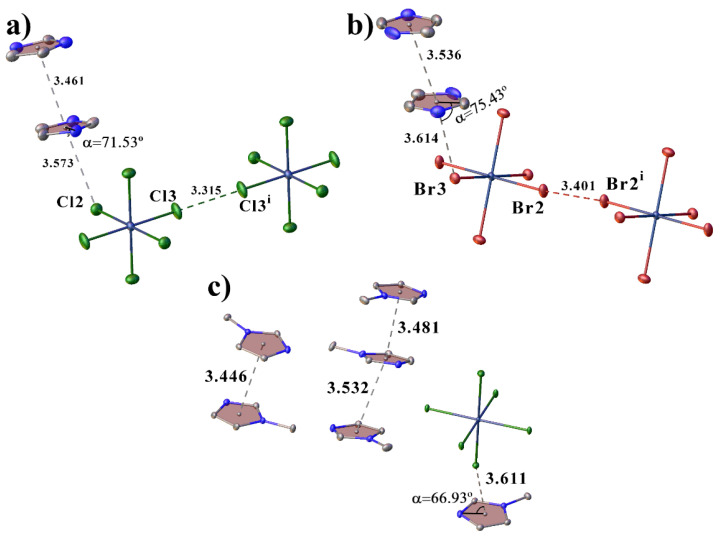
View showing halogen bonds, anion–π and π–stacking interactions in the structures **VI** (**a**), **VII** (**b**) and **VIII** (**c**).

**Figure 7 ijms-23-09461-f007:**
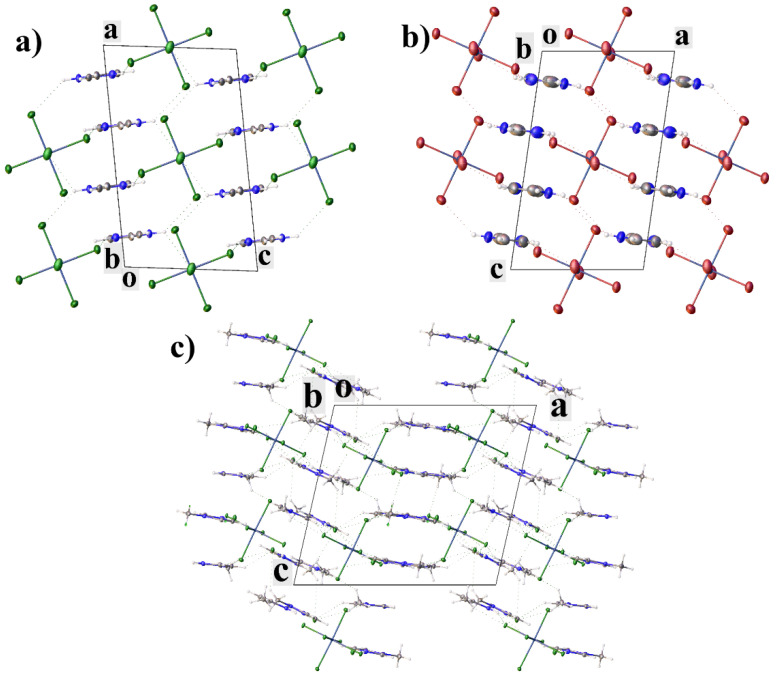
Crystal packing of **VI** (**a**), **VII** (**b**) and **VIII** (**c**), showing cation.

**Figure 8 ijms-23-09461-f008:**
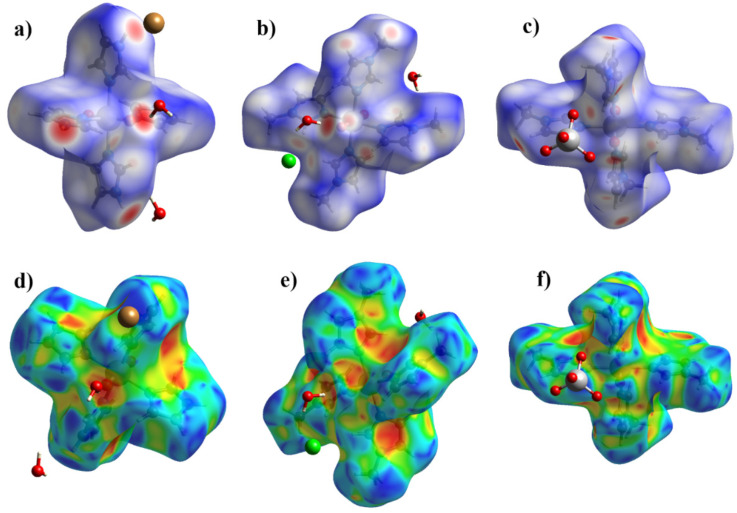
HS mapper over *d*_norm_ for **I** (**a**), **II** (**b**), **III** (**c**) and HS mapper shape-index of **I** (**d**), **II** (**e**), **III** (**f**) to visualize intermolecular interactions in crystals.

**Figure 9 ijms-23-09461-f009:**
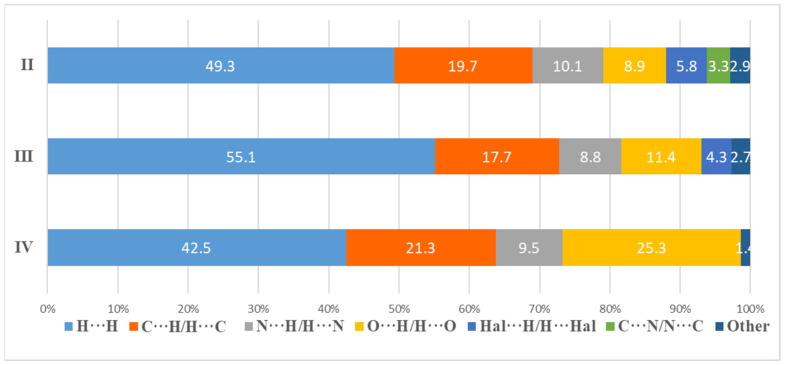
Percentage contributions to the Hirshfeld surface area for the various close intermolecular contacts for **II**–**IV**.

**Figure 10 ijms-23-09461-f010:**
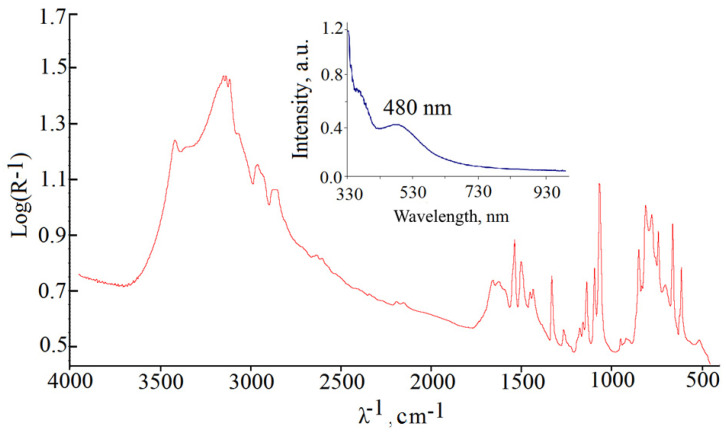
IR spectrum of compound **II**, inset: UV-vis spectroscopy of a methanolic solution of compound **II**.

**Figure 11 ijms-23-09461-f011:**
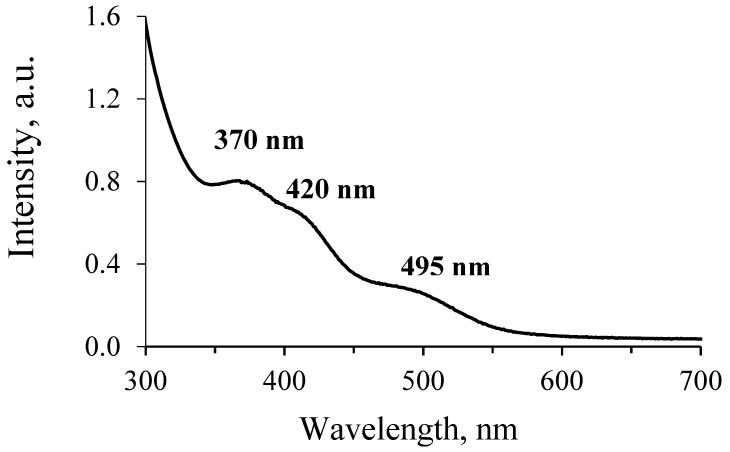
UV-Vis spectroscopy of the stock solution of compound **III** a few minutes after the addition of HTcO_4_.

**Figure 12 ijms-23-09461-f012:**
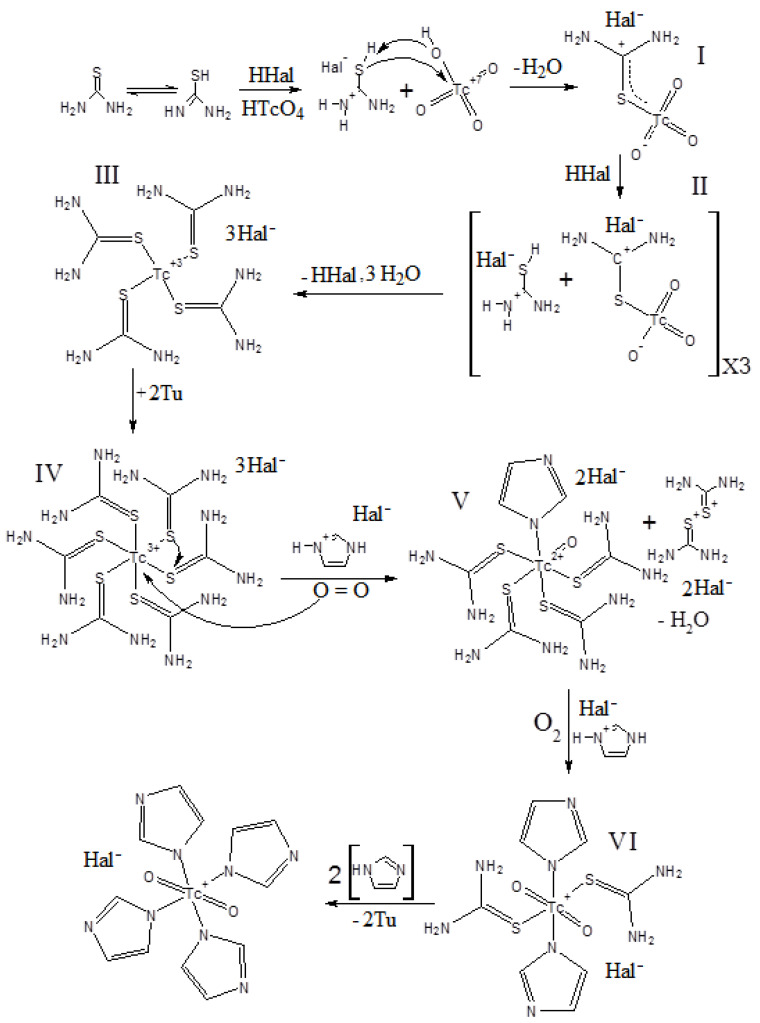
Proposed mechanism of Tc(VII) reduction in the presence of competitive ligands by thiourea with the formation of intermediate products.

**Table 1 ijms-23-09461-t001:** Parameters of π–stacking interactions.

Structure	Rings		Angle	Centroid–Centroid Distance	Shift Distance
**II**	N1C3C2N2C1	N3C5C6N4C4 (symmetry code: −1/2+*x*, 1/2+*y*, *z*)	6.503	3.749	1.175
		N3C5C6N4C4 (symmetry code: −1/2+*x*, 1/2−*y*, −1/2+*z*)	5.351	3.979	1.718
**III**	C10N7C12C11N8	C7N5C8C9N6 (symmetry code: 1+*x*, *y*, *z*)	1.325	3.569	1.269
**V**	C5N1C2N3C4	C15N11C12N13C14 (symmetry code: 1−*x*, 1−*y*, 1−*z*)	4.667	3.437	1.153
	C15N11C12N13C14	C15N11C12N13C14 (symmetry code: 1−*x*, 2−*y*,1−*z*)	0.000	3.342	0.572
**VI**	C1N5C4C3N2	C1N5C4C3N2 (symmetry code: 2−*x*, 1−*y*, 2−*z*)	0.000	3.461	0.937
**VII**	C1N5C4C3N2	C1N5C4C3N2 (symmetry code: −*x*, 1−*y*, −*z*)	0.000	3.536	0.960
**VIII**	C6N2C7C11N1	C17N9C21C19N4 (symmetry code: *x*, 5/2−*y*, −1/2+*z*)	3.045	3.446	1.102
	C13N8C23C20N3	C14N12C18C15N5 (symmetry code: *x*, 1+*y*, *z*)	7.555	3.532	0.892
		C14N12C18C15N5 (symmetry code: *x*, 3/2−*y*, 1/2+*z*)	6.556	3.481	1.136

**Table 2 ijms-23-09461-t002:** Absorption peaks of compound **II** in the IR spectrum.

Peak Position	Intensity	Peak Position	Intensity
661.82	0.915	1264.79	0.689
659.95	1.070	1330.65	0.883
739.93	1.045	1434.10	0.834
775.83	1.104	1500.61	0.934
809.79	1.134	1537.21	1.012
847.68	0.979	1655.87	0.867
1065.58	1.221	2856.37	1.192
1094.15	0.912	2961.85	1.282
1137.11	0.864	3151.49	1.605

**Table 3 ijms-23-09461-t003:** Crystal data and structure refinement for structures **II**–**VIII**.

Identification Code	II	III	IV	V	VI	VII	VIII
Empirical formula	C_12_H_20_BrN_8_O_4_Tc	C_16_H_28_ClN_8_O_4_Tc	C_16_H_24_N_8_O_6_Tc_2_	C_9_H_17_N_6_O_4_STc	C_6_H_10_Cl_6_N_4_Tc	C_6_H_10_Br_6_N_4_Tc	C_16_H_28_Cl_8_N_8_Sn_0.73_Tc_0.27_
Formula weight	518.27	529.91	620.43	403.34	448.88	715.64	729.22
Temperature/K	100 (2)				296 (2)		100 (2)
Crystal system	monoclinic	triclinic	monoclinic	monoclinic	monoclinic	monoclinic	monoclinic
Space group	*Cc*	*P*-1	*C2/c*	*P*2_1_/*n*	C2/*m*	*P2_1_/c*	*P*2_1_/*c*
a/Å	13.0113(11)	8.2324(7)	14.8088(9)	10.5966(3)	12.4407(5)	7.6991(9)	14.7536(5)
b/Å	11.3064(9)	9.4890(8)	13.1343(8)	8.6220(3)	7.9932(4)	8.3979(9)	15.0790(5)
c/Å	14.2554(16)	15.2946(13)	12.9484(8)	17.2476(6)	7.4078(3)	12.8406(16)	13.4854(5)
α/°	90	107.160(3)	90	90	90	90	90
β/°	114.312(4)	95.947(3)	115.911(2)	94.804(1)	97.352(3)	98.166(7)	102.729(1)
γ/°	90	96.731(3)	90	90	90	90	90
Volume/Å^3^	1911.1(3)	1121.63(17)	2265.3(2)	1570.27(9)	730.58(6)	821.81(17)	2926.36(18)
Z	4	2	4	4	2	2	4
ρ_calc_g/cm^3^	1.801	1.569	1.819	1.706	2.041	2.892	1.655
μ/mm^−1^	2.879	0.800	1.270	1.073	2.065	15.447	1.524
F(000)	1032.0	544.0	1240.0	816.0	438.0	654.0	1457.0
Crystal size/mm^3^	0.4 × 0.12 × 0.1	0.3 × 0.27 × 0.25	0.18 × 0.1 × 0.06	0.36 × 0.22 × 0.18	0.4 × 0.23 × 0.19	0.22 × 0.12 × 0.1	0.18 × 0.12 × 0.1
Radiation	MoKα (λ = 0.71073)						
2Θ range for data collection/°	9.316 to 59.996	8.264 to 59.992	8.336 to 59.998	8.374 to 70	8.508 to 49.91	8.204 to 64.974	8.224 to 70
Index ranges	−15 ≤ h ≤ 18, −15 ≤ k ≤ 15, −20 ≤ l ≤ 20	−11 ≤ h ≤ 11, −13 ≤ k ≤ 11, −21 ≤ l ≤ 21	−20 ≤ h ≤ 20, −18 ≤ k ≤ 18, −18 ≤ l ≤ 18	−17 ≤ h ≤ 17, −13 ≤ k ≤ 13, −27 ≤ l ≤ 27	−14 ≤ h ≤ 14, −9 ≤ k ≤ 9, −8 ≤ l ≤ 8	−11 ≤ h ≤ 10, −12 ≤ k ≤ 12, −19 ≤ l ≤ 19	−23 ≤ h ≤ 21, −23 ≤ k ≤ 22, −21 ≤ l ≤ 21
Reflections collected	13375	19898	23979	72747	2927	18418	44345
Independent reflections	4644 [R_int_ = 0.0376, R_sigma_ = 0.0435]	6537 [R_int_ = 0.0425, R_sigma_ = 0.0557]	3283 [R_int_ = 0.0547, R_sigma_ = 0.0371]	6875 [R_int_ = 0.0488, R_sigma_ = 0.0254]	685 [R_int_ = 0.0336, R_sigma_ = 0.0279]	2960 [R_int_ = 0.0400, R_sigma_ = 0.0293]	12760 [R_int_ = 0.0370, R_sigma_ = 0.0389]
Data/restraints/parameters	4644/8/248	6537/0/281	3283/0/149	6875/0/192	685/30/58	2960/0/79	12760/0/315
Goodness-of-fit on *F^2^*	1.030	1.039	1.049	1.051	1.065	1.052	1.020
Final R indexes [I ≥ 2σ (I)]	R_1_ = 0.0223, wR_2_ = 0.0470	R_1_ = 0.0487, wR_2_ = 0.1090	R_1_ = 0.0320, wR_2_ = 0.0676	R_1_ = 0.0238, wR_2_ = 0.0510	R_1_ = 0.0259, wR_2_ = 0.0591	R_1_ = 0.0415, wR_2_ = 0.0877	R_1_ = 0.0296, wR_2_ = 0.0557
Final R indexes (all data)	R_1_ = 0.0235, wR_2_ = 0.0474	R_1_ = 0.0665, wR_2_ = 0.1204	R_1_ = 0.0444, wR_2_ = 0.0729	R_1_ = 0.0310, wR_2_ = 0.0539	R_1_ = 0.0306, wR_2_ = 0.0612	R_1_ = 0.0665, wR_2_ = 0.0966	R_1_ = 0.0477, wR_2_ = 0.0605
Largest diff. peak/hole/e Å^−3^	0.34/−0.37	3.81/−1.88	0.65/−0.90	0.59/−0.51	0.47/−0.46	1.76/−1.03	1.50/−0.51
Flack parameter	0.588(7)						

## Data Availability

The work does not contain information classified as a state secret and can be published in the public domain.

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
