# Peer review of "Novel Synthesis Methods of New Imidazole-Containing Coordination Compounds Tc(IV, V, VII)—Reaction Mechanism, Xrd and Hirshfeld Surface Analysis"

_ijms, 2022, doi:10.3390/ijms23169461_

Round 1

Reviewer 1 Report

This paper refers to the synthesis of novel imidazole-coordinated technetium complexes and their structural and Hirshfeld surface analyses. As mentioned in the introduction, the use of technetium complexes as radiopharmaceuticals in vivo is increasing in importance. Since technetium is a radioactive element, there are fewer papers on research on technetium complexes than on other transition metal complexes. This paper is important as a paper on technetium compounds with several oxidation states. If my comments below are properly corrected, this paper should be published in Int. J. Mol. Sci.

In the introduction and Section 2.1, the authors have compared the structure of the synthesized compound II with the previously reported [TcO2(Im)4]Cl2H2O. The authors do not define compound II to be [TcO2(Im)4]Br2H2O in the text, which makes the description difficult to understand. Similarly, compounds III-VIII must also be defined. Furthermore, ref. 17 also reported the structure of [TcO2(MeIm)4]Cl2H2O. The space group of the compound is C2/c, while the compound III of the same composition in the paper is P-1. The authors should also mention this point.

Furthermore, in Section 2.1, the authors state that molecular ions were packed by π-π stacking between imidazoline rings in compounds II, III, VVIII, and no such interaction was observed in IV. Indeed, in IV, there was no π-π stacking, but the methyl hydrogen of methylimidazole and the CH-π bond of the imidazole ring exist, and the molecular ions are packed. The CH-π bond is one of the factors that determine the molecular structure within the crystal. The authors have to mention the CH-π bond of compound IV.

In Section 2.5 the authors describe the reaction mechanism. The reaction mechanism depicted in Figure 11 must be modified. First, electron arrows are pointing from SH of Tu in the center of the upper row to HO-TcO3, but this is the opposite. The ligand Tu of compounds III-VI was described with a Tc-S-C single bond. The ligands of these compounds are bound as thiocarbonyls (Tc-S=C). Compound III (ref. 21) is an authentic compound and has been done X-ray structural analysis. Next, I cannot understand the transfer row of electrons between compound IV and the imidazolium ion. Is the onium ion a nucleophile? In addition, I did not understand Tu22+. (H2N(H2N=)C-S-S-C(=NH2)NH2)2+?

Minor issues

Line 20: lead -> leads

Line 35: role -> roles

Line 62: Na2S2O3 -> Na2S2O3

Line 73: bounding -> bonding or binding?

        aminoacids -> amino acids

Line 83: its’ -> remove

Line 95: were -> was

Line 98: There are underlines at degree “o”. lines 105~109, 149.

Line 158: second -> the second

Line 171: C2/m -> C2/m

Line 183: orthorombic -> orthorhombic

Lines 229, 231, and Figure 10: sm-1 -> cm-1

Line 358: as inversion -> as an inversion

Line 381: C2/c -> C2/c  

Line 425: V. V. -> V.V.

Line 430: G. V. -> G.V.

Refs. 4, 12, 13, 14: et al. -> all authors

Line 435: K. V. -> K.V.

Refs. 5, 11, 12, 14, 16,17,19, 20, 21, 26, 35, 47: Superscripts and subscripts for Nos. check!

Line 437: Chem. – A Eur. J. -> Chem. Eur. J.

Refs. 9, 37: Check year, Vol. pages.

Line 460: Chemie -> Chem.

Refs. 17, 31, 32, 40: Inorganica -> Inorg.

Line 474: J. Chem. Soc. Chem. Commun. 1994, 0, 2153 -> J. Chem. Soc., Chem. Commun. 1994, 2153.

Line 488: A. V. -> A.V.

Refs. 29, 44: Dalt. Trans. -> Dalton Trans.

Line 43: A. V. and Y. V. -> A.V. and Y.V.

Pages 10 and 13 Table 3 Duplicated

Page 13 Table 3: P21/c ->  P21/c

Reviewer 2 Report

The manuscript deals with some new chemistry of the technetium(V) compounds. Actually, 90% of the manuscript describes crystal structures of target compounds and numerous byproducts, which makes it more suitable for the sister "Crystals" journal. Nevertheless, analysis of crystal structures done at the high level, with highlighting important noncovalent interactions (H-bonds, π-stacking) and supported by Hirshfeld surface analysis. However, that part of manuscript which was done for eligibility for IJMS publishing  - 2.5. Proposed mechanism for the formation of complexes [TcO2(Im)4]+ - looks absolutely artificial. From Figure 11 only two first rows, describing initial reduction by thiourea has very limited sense. All other intermediates are just result of ion exchange, while oxygen appearing from nowhere (indeed from air , but putting all details for thiourea part requires the same detail level for oxygen). The phrase “The synthesis in an oxygen-free atmosphere hindered the formation of the target complexes; Tc(III) compounds could not be isolated” which pretends to be an explanation of the fig 11 is dubious. If authors suppose Tc(III) intermediates in reaction mixture, they should be proved by some methods. E.g. it could be UV-vis, especially since authors mentioned "An intense coloration of solutions", and Tc(III) thiourea complex has the two-peak spectra according to ref 21, which differs from data presented for complex II. And the final moment, the SCXRD typically is not accepted as a single method for the new substance characterization, please provide at least elemental analysis.

Some fount typos:

p2, l85 non-valence interactions - recommended "noncovalent", without hyphen, as IUPAC typically uses it.

P10, L248 TcO2 – make subscript

Remove template fragments from the "Author Contributions" section

P9, L230 : "region of 2800-3700 sm-1, correspond to vibrations of carbon and nitrogen atoms in the imidazole"              this region actually corresponds to the CH, NH, OH (generally XH) vibrations. Also, here and many times more (including axis on fig 10) sm-1 used instead of cm-1

Round 2

Reviewer 2 Report

Scheme of the mechanism became more clear for understanding, but still could be improved by using uniform fonts, size of ligands etc.
Introduced element analysis, which includes technetium, should be accompanied by used techniques in the experimental part.

There is still part of template in the "Author Contributions" section "The following statements should be used" and misprint in author name - "Guerman" (two times).
